# Dispersion Performances of Naphthalimides Doped in Dual Temperature- and pH-Sensitive Poly (N-Isopropylacrylamide-co-acrylic Acid) Shell Assembled with Vinyl-Modified Mesoporous SiO_2_ Core for Fluorescence Cell Imaging

**DOI:** 10.3390/polym15102339

**Published:** 2023-05-17

**Authors:** Xiaohuan Xu, Xiaoli Wang, Xueqing Cui, Bingying Jia, Bang Xu, Jihong Sun

**Affiliations:** Beijing Key Laboratory for Green Catalysis and Separation, Beijing University of Technology, Beijing 100124, China

**Keywords:** naphthalimides, dispersion behavior, stimuli-sensitive composite, fractal features, fluorescent imaging

## Abstract

Developing effective intelligent nanocarriers is highly desirable for fluorescence imaging and therapeutic applications but remains challenging. Using a vinyl-grafted BMMs (bimodal mesoporous SiO_2_ materials) as a core and PAN ((2-aminoethyl)-6-(dimethylamino)-1H-benzo[de]isoquinoline-1,3(2H)-dione))-dispersed dual pH/thermal-sensitive poly(N-isopropylacrylamide-co-acrylic acid) as a shell, PAN@BMMs with strong fluorescence and good dispersibility were prepared. Their mesoporous features and physicochemical properties were extensively characterized via XRD patterns, N_2_ adsorption–desorption analysis, SEM/TEM images, TGA profiles, and FT-IR spectra. In particular, their mass fractal dimension (*d_m_*) features based on SAXS patterns combined with fluorescence spectra were successfully obtained to evaluate the uniformity of the fluorescence dispersions, showing that the *d_m_* values increased from 2.49 to 2.70 with an increase of the AN-additive amount from 0.05 to 1%, along with the red shifting of their fluorescent emission wavelength from 471 to 488 nm. The composite (PAN@BMMs-I-0.1) presented a densification trend and a slight decrease in peak (490 nm) intensity during the shrinking process. Its fluorescent decay profiles confirmed two fluorescence lifetimes of 3.59 and 10.62 ns. The low cytotoxicity obtained via in vitro cell survival assay and the efficient green imaging performed via HeLa cell internalization suggested that the smart PAN@BMM composites are potential carriers for in vivo imaging and therapy.

## 1. Introduction

Stimulus-responsive inorganic–organic nanocomposites have been widely used in controlled drug delivery systems to target the lesion site, which could be beneficial to achieving a higher efficacy of chemotherapeutic drugs and reducing the side effects on normal healthy tissues [1,2,3]. Nowadays, the core–shell strategy is especially useful in designing a stimulus-responsive drug carrier with a core of mesoporous silica nanoparticles (MSNs) and a stimuli-responsive polymer shell [4,5]. For example, Yang et al. [6] reported that the pH and thermal dual stimuli-responsive drug carriers constructed from the shell of P(MAA-co-NIPAM) copolymer and the core of Fe_3_O_4_/mSiO_2_, in which, methacrylic acid (MAA) played the role of pH response and N-isopropylacrylamide (NIPAM) played the role of temperature response. Using doxorubicin hydrochloride as a model drug, the resultant P(MAA-co-NIPAM)@Fe_3_O_4_/mSiO_2_ exhibited a high drug loading of 114.32 mg/g (11.43%) at 25 °C and a maximum drug release rate of 93.94% at pH 6.0, showing an excellent controlled-release behavior. Obviously, the hybridizing MSNs coated with an intelligent copolymer have the rapid response ability of the polymer gatekeeper to external/inside stimuli and the excellent drug-loading capacity of the MSN carrier, owing to their high specific surface area and large pore volume [7].

Specifically, temperature and pH are commonly used as stimuli-response signals for favored drug release due to variation of the microenvironment between different tissues or parts in the human body [8]. Recently, Liu et al. [9] reported the synthesis of a pH/thermal-triggered nanocomposite with double-bond-modified MSNs as a core and P(NIPAM-co-MAA) as a shell, and fluorescein was selected as a tracer.

The results verified that the P(NIPAM-co-MAA) shell of the obtained composite nanocarrier could be network-open under a low pH (3.5) and high temperature (45 °C) condition; in this way, the 60.3 wt% fluorescein was controllably released from mesopores of the used MSNs. Compared with conventional MSNs, Jin et al. in our group [10,11,12] constructed a variety of core–shell structured composite nanoparticles based on bimodal mesoporous SiO_2_ materials (BMMs) as a core and the dual (thermo- and pH-)-responsive P(NIPAM-co-AA) as a shell. Using ibuprofen (IBU) as a model drug, the release profiles of the P(NIPAM-co-AA)@BMMs demonstrated that the shrinking performance of the coated copolymer shell could effectively moderate the total diffusion, reaching 95% of the IBU-releasing rate at 37 °C/pH 2.0, higher than the release amount of 33% at 37 °C/pH 7.4. In particular, the microstructures of the coated copolymer shell, including coating thicknesses and fractal features and their evolutions during the IBU-loading or releasing behaviors, were evaluated via small-angle X-ray scattering (SAXS) patterns.

However, the desired position of the drugs and the reasonable timing of the release behaviors in vivo are difficult to detect [13]. For this reason, drug carriers doped with fluorescent molecules have emerged as a powerful tool, which has progressively been developed to meet the pressing clinical need for fluorescent imaging and drug tracking [14,15].

Accordingly, a strong and high photochemical stability is crucial; besides this, a low cytotoxicity is usually required [16]. For instance, Zhao et al. [17] prepared Eu-doped PS-co-PNIPAM/d-TPE-doped PNIPAM-co-PAA hydrogel nanoparticles, in which the thermo- and pH-responsive copolymers of d-TPE-doped PNIPAM-co-PAA acted as the smart shell and the Eu-doped PS-co-PNIPAM acted as a core. The photoluminescence intensities of the nanoparticles exhibited a red emission (613 nm) from the Eu-doped core and a blue emission (468 nm) from the d-TPE-doped shell at similar excitation wavelengths (360 nm), making the pH- and temperature-responsive nanohydrogel advantageous for the fluorescent probe. More importantly, the cell viability of Eu-doped PS-co-PNIPAM/d-TPE-doped PNIPAM-co-PAA nanoparticles remains above 80%, presenting good biocompatibility.

However, the short emission wavelength and small Stokes shifts of these fluorescent molecules severely limited their potential [18]. In this regard, fabricating near-infrared (NIR) absorbing or emitting wavelength materials with superior optical penetration and lesser photodamage is vitally important in clinical applications [19,20]. The introduction strategy of electron-donor units (amine, hydroxyl, alkoxy, or aryl groups) is generally designed to regulate fluorescent behaviors such as wavelength, color, lifetime, and quantum yield [21]. Compared with the traditional organic fluorescent molecules, 1,8-naphthalic anhydride (NA) with a strong fluorophore group is an attractive probe because its luminescent properties can be altered by the introduction of various electron-donating or electron-accepting substituents (D-π-A) at the 4- position or the naphthylamide ring [22,23].

Recently, Cao et al. [24] synthesized the 1,8-naphthalimide derivatives for visualizing rRNA in live cells. They identified the probe 1 (2-(2-aminoethyl)-6-(dimethylamino)-1H-benzo[de]isoquinoline-1,3(2H)-dione, abbreviated as AN, in which the dimethyl amine group acted as an electron donor, whereas the 1,8-dicarboxyl groups acted as electron acceptors. Their photoluminescence (PL) spectra results showed a maximum fluorescent emission wavelength of 545 nm at 40 μM in a phosphate buffer solution (PBS) at pH 7.2, exhibiting a large Stokes shift of 101 nm. Meanwhile, the intracellular imaging experiment presented a bright green fluorescence (515–565 nm) and a high selectivity towards rRNA. Based on this, the excellent features of AN encouraged us to explore its luminescence mechanism in a hybrid nanocarrier.

In our preliminary work [25,26,27], a series of luminescent hybrid nanoparticles with effectively luminescent performances was prepared using BMMs as a carrier and NA derivatives as luminescent probes through surface modification, presenting a longer fluorescent lifetime (2.2–3.4 ns) that was caused by the delocalized pair of electrons in the electron-withdrawing conjugate structures. Thereafter, Liu et al. and Wei et al. [28,29] further developed a series of core–shell structured nanocomposites with strong fluorescent features and dual temperature- and pH-responsive properties through self-assembly technology, in which the BMM was used as a core and the fluorescent molecule PID-dispersed P(NIPAM-co-AA) acted as a shell. Notably, the resultant hybrid nanocomposite PID/P@BMMs with two lifetimes of 2.53 and 18.86 ns not only possessed an excellent controlled IBU release of 71.1% at 37 °C and 82.7% at pH 2.0 but also presented a stable fluorescence intensity at the 395–450 nm wavelength during drug loading and releasing. In addition, the “surface effect” and “porous confined effect” on the luminescent behaviors were preliminarily investigated [30].

Nevertheless, few reports focused on the relationship between the luminescent properties and their dispersion behaviors in hybrid composites have been published over the past few years. In essence, an aggregation state of the fluorescent molecules affects their luminescent performances (high efficiency, sensitivity, strong fluorescence) as tracers [31]. Accordingly, one of the main goals in the present work is to explore the dispersion behaviors of the fluorescent molecular AN in the core–shell structured composites for achieving an excellent fluorescent property and the stimuli-responsive for further research of drug nanocarrier.

The SAXS technique is a powerful method for analyzing the representative microstructural information (such as the fractal dimension, the characteristic shape and size, and the thickness of the interfacial layer) of various porous materials or macromolecule copolymers. Xie et al. [32] used the SAXS method to elucidate the nanoporous structures of non-caking coal briquettes during carbonization, clearly demonstrating the evolution of pore physical structures. Based on these encouraging explorations, the SAXS method combined with other characterizations are used for demonstrating the microstructural properties of dual pH- and thermal- sensitive hybrid composites in the present work. In this work, not only their fractal dimensions during swelling or shrinking were verified, deriving from their integrated scattering patterns, but also their particle morphologies and shape geometries were estimated on the basis of the pair distance distribution function (*P(r) ~ r*) (PDDF) profiles. Obviously, the novelty of this investigation is that the feasibility of preparing core–shell structured composites was specifically demonstrated by a fractal dimension combined with other characterizations.

Herein, using the vinyl-grafted BMM as a core and the uniformly AN-dispersed P(NIPAM-co-AA) as a shell (abbreviated as PAN), we designed out a series of dual temperature- and pH-responsive fluorescent nanoparticles (PAN@BMMs) via a radical polymerization approach. The fluorescent performances for their dual temperature- and pH-responsive properties were discussed, and the dispersive behaviors of AN doped in P(NIPAM-co-AA) networks, such as the AN-doped way and dosage, the swelling or shrinking kinetics as a function of emission wavelength, as well as fluorescent intensity and decay lifetime, were emphatically discussed. In particular, the fractal evolutions of doped AN in PAN@BMMs-m-n during the swelling and shrinking process under various conditions were further demonstrated by SAXS patterns on the fractal dimensions theory to clarify the essence luminescence mechanism and evaluate the drug release performance. Moreover, the structural features and textural parameters of the fluorescent hybrid nanocomposite PAN@BMMs-m-n were extensively characterized with other various techniques, such as X-ray diffraction (XRD) patterns, scanning electron microscope (SEM) and transmission electron microscope (TEM) images, N_2_ sorption isotherms, Fourier transform infrared (FT-IR) spectra, and thermogravimetric analysis (TG). In addition, the low critical solution temperature (LCST) of representative fluorescent drug carriers was confirmed by differential scanning calorimetry (DSC) measurements. Meanwhile, their dual temperature- and pH- sensitivities were demonstrated through dynamic light scattering (DLS) measurement and hydrodynamic diameter (*D_h_*) analysis under various environmental conditions. Finally, the cytotoxicity of these nanocomposites was tested in vitro for HeLa cancer cells. Their uptake process and distribution in the cell were imaged using fluorescence imaging.

## 2. Materials and Methods

### 2.1. Materials

N-isopropyl acrylamide (NIPAM, 98%, purified by recrystallization in n-hexane), acrylic acid (AA, A.R.), N, N′-methylene bisacrylamide (BIS, A.R.), potassium persulfate (KPS, A.R.), sodium dodecyl sulfate (SDS, A.R.), 3-methacryloxypropyltri-methoxysilane (MPS, A.R.), 1,4-Dioxane, acetonitrile (MeCN), tetraethyl orthosilicate (TEOS, A.R.), and cetyltrimethylammonium bromide (CTAB, A.R.) were purchased from Aladdin Chemistry Co., Ltd., (Shanghai, China). (2-(2-aminoethyl)-6-(dimethylamino)-1H-benzo[de]isoquinoline-1,3(2H)-dione) (AN) was obtained from Shanghai Haohong Bio-Pharm Technology Co., Ltd. (Shanghai, China). Ammonium hydroxide (NH_4_OH, 25%), ammonium nitrate (NH_4_NO_3_, A.R.), petroleum ether (PE, A.R.), n-hexane (A.R.), ethanol (EtOH, A.R.), N, N-dimethylformamide (DMF, A.R.), dimethyl sulfoxide (DMSO, A.R.), methanol (CH_3_OH, A.R.), acetonitrile (C_2_H_3_N, A.R.), and acetone (C_3_H_6_O, A.R.) were provided by Tianjin Fuchen chemical reagents factory. Disodium hydrogen phosphate, sodium dihydrogen phosphate, and phosphoric acid were supplied by Beijing chemical works. Human cervical cancer cells (HeLa) were obtained from Saiqi Biological Engineering Co., Ltd (Shanghai, China). Glutamine, Dulbecco’s modified Eagle’s medium (DMEM) cell culture medium, fetal bovine serum (FBS), penicillin/streptomycin, 0.25% trypsin/EDTA solution, and phosphate-buffered saline (PBS) were obtained from HyClone. Cell Counting Kit-8 (CCK-8) was supplied by Dojindo Molecular Technologies (Dojindo, Kumamoto, Japan). Ultra-pure water (18.2 MΩ cm^−1^, Millipore Co., Burlington, MA, USA) was used in all experiments. PBS for the drug-releasing experiment was prepared as follows: Solution A: 1.66 mL of phosphoric acid was measured and then adjusted to 1000 mL with deionized water. Solution B: 7.16 g of disodium hydrogen phosphate was weighed, and then its volume was made up to 1000 mL by adding deionized water. Solution C: 3.12 g of sodium dihydrogen phosphate was used, and then deionized water was added to make its volume up to 1000 mL. PBS at pH 2.0 was prepared by mixing 725 mL of Solution A and 275 mL of Solution B. PBS at pH 7.4 was prepared by mixing 810 mL of Solution B and 190 mL of Solution C.

### 2.2. Preparation and Double-Bond Modification of t-BMMs

The t-BMMs in the presence of CTAB were synthesized partially according to the method reported in the literature [33]. In detail, under continuous magnetic stirring, 20.9 g of CTAB was dissolved in 832 mL of deionized water, followed by 64 mL of TEOS added dropwise. Then, 19.2 mL of ammonia was added quickly. The white gel obtained from the above reaction mixture was washed with plenty of water, dried at 120 °C for 3 h, and encoded as t-BMMs.

In a typical modifying procedure, MPS with a double bond was further modified on the surface of t-BMMs before removing the CTAB surfactant. t-BMMs (3.5 g) were dispersed into an anhydrous ethanol solution of 1% MPS (350 mL). After stirring at 60 °C for 4 h, the mixture was filtered and repeatedly washed with ethanol. The M-t-BMMs were obtained after drying at 60 °C in a vacuum oven (−0.08 MPa) for 12 h.

### 2.3. Synthesis of PAN@M-t-BMMs-m-n

We ensured that the P(NIPAM-co-AA) polymers were grafted to the vinyl-modified surfaces of the prepared t-BMMs via radical polymerization, in which the formations of the covalent bonds between P(NIPAM-co-AA) and vinyl groups appeared. Meanwhile, interaction forces between the –CO–NH–, –COOH groups distributed in the P(NIPAM-co-AA) polymers and the –NH_2_ groups located in the dye (AN) appeared in the prepared PAN and PAN@BMMs, in which the dispersed or aggregated states of the AN used during the process of swelling–shrinking and drug loading–releasing were conducted. In summary, the preparation process of the core–shell structured I/PAN@BMMs nanoparticles is illustrated in Figure 1.

A series of PAN@M-t-BMMs-m-n hybrid nanocomposites were prepared, where m represents how the fluorescent AN was doped, either with a one pot process (I), a two-step process (II), or by physical mixing (mix); n is the feeding ratio of the mass of AN to the total mass of NIPAM, AA, and M-t-BMMs, equal to 0.05:100 (0.05%), 0.1:100 (0.1%), and 1:100 (1%), respectively. In addition, the feeding mass ratio of AA to NIPAM was 7: 100.

The specific synthesis process was as follows: PAN@M-t-BMMs-I-0.05, -0.1, and -1 were synthesized by a one-pot process: M-t-BMMs (200 mg), NIPAM (40 mg), AA (2.8 μL), BIS (7.3 mg), and a certain amount of AN were dissolved in 200 mL of H_2_O and bubbled through nitrogen for 1 h. Then, the initiator KPS was rapidly added, and the polymerization reaction was kept for 7 h at 70 °C under N_2_. The P(NIPAM-co-AA)@M-t-BMMs nanocomposites and P(NIPAM-co-AA) polymer were synthesized similar to the preparation procedure above. 

PAN@M-t-BMMs-II-0.1 was synthesized by a two-step process as follows: The fluorescent polymer PAN with 0.1% of AN was firstly synthesized via the above copolymerization without M-t-BMMs. In detail, 40 mg of NIPAM, 2.8 μL of AA, and 0.25 mg of AN were dissolved in 200 mL of H_2_O and bubbled through N_2_ for 1 h. Then, the initiator KPS (108 mg) was rapidly added, and the polymerization reaction was kept for 7 h at 70 °C under N_2_. Then, 0.4 g of the prepared fluorescent polymer PAN and 1.0 g of M-t-BMMs were dispersed into 50 mL of ethanol and refluxed with continuous stirring for 7 h at 70 °C.

Thirdly, the PAN@M-t-BMMs-mix-0.1 (0.1 was the mass ratio (0.1 wt%) of AN/P(NIPAM-co-AA) with M-t-BMMs) was obtained by physically mixing 1 g of P@M-t-BMMs with 10 mL of AN-contained aqueous solution (0.04 mg/mL) and then dried at 80 °C for 12 h.

### 2.4. Removing CTAB

About 1 g of t-BMMs, M-t-BMMs, P(NIPAM-co-AA)@M-t-BMMs, or PAN@M-t-BMMs-m-n was extracted in 100 mL of ammonium nitrate ethanol solution (10 mg/mL) at 80 °C with reflux for 12 h, repeated three times. The resulting solid particles were named BMMs, M-BMMs, P(NIPAM-co-AA)@BMMs, and PAN@BMMs-m-n after centrifuging.

### 2.5. Temperature/pH Sensitivity Tests and the Dispersive Behavior of Fluorescent AN

The sensitivity measurement of the hybrid nanoparticles PAN@BMMs-m-n was performed using DLS analysis. Generally, 10 mg of the sample was immersed in 10 mL of PBS solution (pH 7.4 or pH 2.0) at 25 °C or 37 °C. After it reached the swelling equilibrium, the *D_h_* value of the PAN@BMMs-m-n particles was examined.

Simultaneously, the variation of the fluorescence emission spectra of the PAN@BMMs-m-n along with swelling time was monitored in the following description test process: at a particular interval until equilibrium (1, 3, 8, 12, and 24 h), the solid particles and corresponding filtrates were obtained by centrifugation, and then their PL spectra were recorded using a fluorescence spectrometer.

### 2.6. Drug Loading and Release Performances

The resultants of PAN@BMMs-m-n (0.2 g) were dispersed into 20 mL of IBU solution in anhydrous ethanol (20 mg/mL) and were stirred at room temperature for 48 h. The IBU-loaded samples were filtered and washed with anhydrous ethanol and then dried at 80 °C for 12 h. To measure the loading amount of IBU, the filtrate was diluted up to 50 mL and then analyzed using high-performance liquid chromatography (HPLC) (the standard calibration curve is shown in Appendix A). These IBU-loaded samples were named I/PAN@BMMs-m-n.

The formula used for calculating drug-loading capacity (*LC%*) was
(1)$$LC\%=\frac{m_{0}-m_{1}}{m_{2}}\times 100\%$$
where *m*_0_ is the total weight of IBU initially, *m*_1_ is the weight of IBU remaining in the soaked medium after equilibrium, and *m*_2_ is the weight of I/PAN@BMMs-m-n.

In vitro drug release was studied in 100 mL of PBS solution (pH 2.0 or 7.4) incubated in a shaking water bath at 25 °C or 37 °C. Generally, about 25 mg of the drug-loaded sample was put into the dialysis bag (molecular weight cutoff: 3500 Da), which was immersed in 20 mL of the solution. At a particular interval, 1 mL of the released solution was withdrawn and, at the same time, 1 mL of fresh solution was supplemented to keep the pH value of the drug-release system unchanged. The concentration of IBU released from the sample was analyzed using a UV-visible spectrophotometer at a wavelength of 272 nm (the standard calibration curves are shown in Appendix A). The drug release content was calculated according to the following Formula (2):(2)$$Cumulative\;Releasepercent\;\%=\sum_{t=0}^{t}\frac{C_{t}\times V}{M}=100\%$$
where *C_t_* (mg/mL) is the apparent concentration at time *t*, *V* is the volume of total volume of release fluids (mL), and *M* is the IBU loading amount.

### 2.7. Cytotoxicity Assay

The in vitro cytotoxicity of the nanocomposite carrier (BMMs, P(NIPAM-co-AA), AN, PAN@BMMs-I-0.05, -0.1, -1) was evaluated using the CCK-8 assay. HeLa cells were passaged three times and then used as the tumor cell models. DMEM was used as the control group. Experiments of each carrier were conducted in triplicate. The detailed procedure was as follows: HeLa cells were cultured in DMEM with 10 vol.% FBS for 48 h; the incubator was maintained at 37 °C with saturated humidity and 5 vol.% CO_2_. The cells were transferred from the culture bottle to 96-well plates at a density of 8 × 10^3^ cells per well. After being cultured for 24 h, the medium was sucked out and 200 µL of the culture medium containing different concentrations (1, 5, 10, 20, 50, 100, 200 μg/mL) of the nanocomposites was added.

After being cultured for 48 h, the culture medium was removed, and the adhered cells were washed with PBS solution. A total of 100 µL of the CCK-8 solution (10 vol.% CCK-8) was then added to each well in the dark. After 1 h of continuous culture in the incubator, the optical density (OD) value at 450 nm was determined using a microplate reader (Thermo Scientific, Waltham, MA, USA). Cell viability values were obtained after data treatment. The *cell viability* values were calculated using Equation (3) as follows:(3)$$Cell\;Viability\left(\%\right)=\frac{{OD}_{Sample}-{OD}_{Blank}}{{OD}_{Control}-{OD}_{Blank}}\times 100\%$$

The OD_Sample_ was the cell cultured in the medium containing samples, the OD_Control_ was the cell cultured in the medium, and the OD_Blank_ was the well with the medium.

Statistical analysis: Statistical analyses were conducted using Excel software. The data of measurement indicators are expressed as mean ± standard deviation. The differences between two groups were analyzed using the unpaired *t*-test. One-way analysis of variance was used to compare multiple groups, followed by the least significant difference in multiple comparisons. *p* < 0.05 was considered statistically significant.

### 2.8. Cell Uptake and Intracellular Distribution

Cellular uptake and the intracellular distributions of the PAN@BMMs-I-0.1 carrier were investigated according to a reported method with slight modification [34]. Briefly, the HeLa cells were incubated in a medium containing PAN@BMMs-I-0.1 (concentration 20 μg/mL). The HeLa cells were adhered to a glass-bottom Petri dish (3 × 10^4^/well, 2 mL cell suspension). After 6 h, 12 h, and 24 h, the cells were washed with PBS three times to remove the excess carrier, and then the intracellular distribution was observed under a confocal laser scanning microscope (CLSM).

### 2.9. Characterizations

The SAXS patterns were carried out using synchrotron radiation at the 1W2A station of the Beijing Synchrotron Radiation Facility [35]. The wavelength of the incident X-ray was 0.154 nm. The sample-to-detector distance for the SAXS was 1590 mm, which was calibrated with the diffraction ring of a standard sample. The scattering vector magnitude *q* ranged from 0.08 to 3.05 nm^−1^. The sample was loaded into a sample cell and sealed with scotch tape on a groove. The thickness of the sample cell was approximately 1 mm. The scattering images were collected through the single-frame mode with an exposure time of 30 s and a ‘multi-read’ of 2 times. The two-dimensional SAXS data were converted to the one-dimensional intensity *I(q)* as a function of *q* (*q* = 4πsinθ/λ) using the Fit2D software (http://www.esrf.eu/computing/scientific/FIT2D, accessed on 25 January 2016) [36] and further processed with the S program package [37]. In this study, the structure information obtained from the mass fractals (*d_m_*, 1 < *α* < 3) or surface fractals (*d_s_*, 2 < 6 − *α* < 3) was calculated according to the power law (*I(q)* = *I_0_q*^−*α*^) [38].

The XRD spectrometry was performed on an XD6 X-ray polycrystal diffractometer (Beijing Purkinje General Instrument Co., Ltd., Beijing, China) using Cu-Kα radiation (*λ* = 0.154056 nm) with a scanning speed of 1°/min at 36 kV and 20 mA. The SEM images were acquired using a Hitachi S-4800 field-emission scanning electron microscope by an Everhart–Thornley detector at a working distance of 10.4 mm; the acceleration voltage was 15.0 kV, and the beam current was 0.34 nA. Before measurement, the hybrid samples were dried and their surfaces were sprayed with a gold spraying current of 10 mA for 30–60 s. The TEM micrographs were collected using a JEOL JEM2100F at 300 kV. The TGA profiles were measured using a thermogravimetric analyzer STA-8000 (PerkinElmer Ltd. Buckinghamshire, UK) in the range of 30–900 °C under an air atmosphere with a heating rate of 10 °C/min. The FT-IR spectra with the range of 4000–400 cm^−1^ were recorded using a PerkinElmer Spectrum 100 spectrometer via KBr tableting. The scans number was 32, and the resolution step was 1 cm^−1^. X-ray Photoelectron Spectroscopy (XPS) characterization was performed using an XPS (Thermo Fisher Scientific ESCALAB 250Xi, Waltham, MA, USA), and the data were analyzed and processed using XPS Peak software. The binding energies of all samples were corrected with a C signal of 284.8 eV as a standard. The textural properties, including surface area and porosity, were acquired from a nitrogen adsorption–desorption isotherm obtained (at 77 K) using a JWGB JW-BK 300 sorption meter. A series of PAN@BMMs hybrid nanocomposites were pretreated at 80 °C for 5 h under a helium atmosphere. The specific surface areas of related samples were provided using the Brunauer–Emmett–Teller (BET) method based on N_2_ adsorption–desorption isotherms. The pore volumes and pore size distributions were measured using adsorption branches of the isotherm measured using the Barrett–Joyner–Halenda (BJH) model. The LCST of the obtained samples was determined on a NETZSCH DSC-214 instrument. The swollen sample was sealed in an aluminum crucible under an N_2_ atmosphere, with a flow rate of 40 mL/min, a test temperature range of 25–50 °C, and a heating rate was 1 K/min. Then, their enthalpy changes (Δ*H*) were calculated using software from NEIZSCH. The *D_h_* and the zeta potential of the hybrid nanoparticles were confirmed by dynamic light scattering measurements on a Malvern Zetasizer Nano ZS90. The fluorescence spectra were recorded using a Hitachi F-7000 fluorescence spectrophotometer with a scan speed of 2400 nm/min and a scanning step length of 1 nm. UV-vis absorbance spectra were measured and recorded using a Shimadzu UV-2600 spectrophotometer. The cells were counted using a cell counting chamber under the inverted microscope (XDS-1B, COIC). The absorbance of plates was measured using a microplate reader (Thermo Scientific, USA) at 450 nm. The cell uptake and PAN@BMMs-I-0.1 distribution in cells were observed using a CLSM (A1Rsi, Nikon) at the excitation source of 488 nm. Time-resolved fluorescence decay was analyzed using the Edinburgh instrument (FLS-1000). A xenon (Xe) lamp (300 nm) was used as the light source. The fitted lifetimes (*τ*_1_, *τ*_2_) and the confidence factor (*χ^2^*) were obtained using the following double exponential function Equation (4):(4)$$R\left(t\right)={A+B}_{1}e^{\left(-\frac{t}{\tau 1}\right)}+B_{2}e^{\left(-\frac{t}{\tau 2}\right)}$$
where *B*_1_ and *B*_2_ are the corresponding percentage of the fitted lifetimes *τ*_1_ and *τ*_2_, and *A* is the constant. 0.8 < *χ^2^* < 1.3 indicates that the fitting results are reliable. The resultant decay curves were fitted in the software provided by the instrument.

## 3. Results and Discussion

### 3.1. Structural and Textural Characterizations

Figure 1A shows the XRD patterns for all related samples in the *2θ* range of 1–10°. Their texture parameters are summarized in Table 1. As shown in Figure 1A (a), the XRD patterns of the BMMs presented a distinctive diffraction peak (100) at *2θ* of 1.92°, indicating a characteristic of typical BMMs with ordered mesopores channels [39]. After vinyl functionalizing by MPS, the diffraction peak of the M-BMMs (Figure 1A (b)) moved slightly toward a larger angle region (2.01°) compared to that of BMMs (Figure 1A (a)), and the corresponding *d* values decreased from 4.5959 nm for BMMs to 4.3900 nm for M-BMMs (as shown in Table 1), implying that the mesopore structures could remain intact after vinyl-modifying, but the interplanar spacing had shrunk [25,40].

Thereafter, the peak intensity of the synthesized fluorescent PAN@BMMs samples (Figure 1A (c) to (e)) seemed to be decreased, lower than that of M-BMMs (Figure 1A (b)), probably suggesting the successful encapsulations of the AN-dispersed polymer onto the surfaces of M-BMMs [41]. Additionally, their peak_(100)_ position was also shifted to the larger angle regions, such as 2.03° for PAN@BMMs-I-0.05 (Figure 1A (c)), 2.02° for PAN@BMMs-I-0.1 (Figure 1A (d)), and 2.06° for PAN@BMMs-I-1 (Figure 1A (e)), accompanied by typically lower *d*_(100)_ values, such as 4.3468, 4.3683, and 4.2835 nm (as shown in Table 1), respectively. One of the main reasons for this was the partial introductions of PAN onto the mesoporous channels of M-BMMs.

After IBU loading, the diffraction peak_(100)_ position of I/PAN@BMMs moved to 1.86° for I/PAN@BMMs-I-0.05 (Figure 1A (f)), 1.82° for I/PAN@BMMs-I-0.1 (Figure 1A (g)), and 1.88° for I/PAN@BMMs-I-1 (Figure 1A (h)), respectively. Accordingly, the corresponding *d*_(100)_ space values increased to 4.7440, 4.8483, and 4.6937 nm (as shown in Table 1), respectively. These observations might be responsible for the IBU loading into the mesoporous channels.

Figure 1A (i) insert displayed a characteristic diffraction peak of the crystal IBU in wide-angle (10–50°) XRD patterns. Nevertheless, I/PAN@BMMs (inset of Figure 1A (f) to (h)) displayed a wide diffraction peak in the 2-theta range of 20–30°, indicating the homogeneous distribution of the loaded IBU. Additionally, similar XRD patterns (Appendix A) were obtained for other PAN-coated and drug-loaded samples synthesized via the two-step and physical mixing methods. The characteristic diffraction peak_(100)_ position moved from 2.02 and 2.06° for PNA@BMMs-II-0.1 and PNA@BMMs-mix-0.1 to 1.87 and 1.96° for IBU-loaded samples (I/PNA@BMMs-II-0.1 and I/PNA@BMMs-mix-0.1), and the corresponding *d*_(100)_ values increased from 4.3683 and 4.2835 nm to 4.7187 and 4.5021 nm. These observations further proved the drug loading and that the PAN copolymers were successfully coated onto the surfaces of the BMM cores. These speculations could be further supported by the N_2_ sorption isotherms.

Appendix A illustrated N_2_ adsorption–desorption isotherms and the corresponding pore size distribution profiles of all related samples. As shown in Appendix A, the isotherms of M-BMMs (Appendix A) and PAN@BMMs (Appendix A) were apparently identified as type IV isotherms with H1 type hysteresis loops, similar to a typical feature of BMMs (Appendix A), implying the presence of the uniform mesopores [42]. Meanwhile, BMMs (Appendix A) exhibited the bimodal mesopore distributions: the first one was centered at 3.12 nm, and the broader second one was about 40.1 nm. The specific surface area of the M-BMMs (Appendix A) decreased slightly from 1132.8 m^2^/g (BMMs) to 1123.9 m^2^/g at a mean pore size of around 3.13 nm and 36.0 nm. These results manifested that the mesoporous structures of M-BMMs modified by organic groups on the outer surface of BMMs remained intact, similar to our previous report [28].

Meanwhile, the specific surface area and cumulative pore volume were found to be 1123.9 m^2^/g and 1.82 cm^3^/g for M-BMMs (Appendix A), almost the same as the measurements for BMMs, which were 1132.8 m^2^/g and 1.89 cm^3^/g (Appendix A), but decreased to 722.9 m^2^/g and 1.13 cm^3^/g for PAN@BMMs-I-0.05 (Appendix A), 738.6 m^2^/g and 0.86 cm^3^/g for PAN@BMMs-I-0.1 (Appendix A), and 799.4 m^2^/g and 0.87 cm^3^/g for PAN@BMMs-I-1 (Appendix A), respectively. These results suggested that on the hand, the vinyl groups were successfully grafted onto the outer surfaces of the BMMs, but on the other hand, the PAN shell coated on the outer surfaces of BMMs impedes the N_2_ adsorption, resulting in a decrease in the adsorption amount and surface areas [11]. Correspondingly, the small pore diameter was observed to slightly decrease from 3.13 for M-BMMs to 2.42−2.92 nm for PAN@BMMs. As expected, the core–shell structured PAN@BMMs with the retained mesoporosity had enough drug loading for a stimulus-responsive release carrier. Subsequently, after IBU loading, their surface areas abruptly declined to around 601.0 m^2^/g for I/PAN@BMMs-I-0.05 (Appendix A), 616.5 m^2^/g for I/PAN@BMMs-I-0.1 (Appendix A), and 778.0 m^2^/g for I/PAN@BMMs-I-1 (Appendix A), respectively. In particular, the small mesopore size decreased accordingly. These phenomena further confirmed the successful IBU encapsulating inside the mesoporous channels of BMM cores. The N_2_ sorption isotherms of PAN@BMMs-II-0.1 (Appendix A) and PAN@BMMs-mix-0.1 (Appendix A) also presented similar results.

In order to determine the temperature response of the PAN@BMMs, their DSC profiles were measured after swelling at room temperature for 24 h. As can be seen in Figure 1B (a), no endothermic peak appeared in the DSC curve of pure BMMs, suggesting that the core of hybrid materials does not possess thermal responsiveness. However, after coating the fluorescent PAN on the outer surfaces of the BMM cores, the LCST was 31.1 °C for PAN@BMMs-I-0.1 (Figure 1B (b)) and 29.3 °C for I/PAN@BMMs-I-0.1 (Figure 1B (c)), almost same as the LCST (32 °C) of pure PNIPAM in the reported literature [43]. As shown in Figure 1B (d) and (e), the LCST of PAN@BMMs-I-0.05 and PAN@BMMs-I-1 varied at 28.5 °C and 31.3 °C, implying that the AN doping slightly affected the LCST of PAN@BMMs.

On the basis of the calculations derived from the NETZSCH software, the enthalpy change (Δ*H*) values exhibited a decreasing trend, 0.87 J/g for PAN@BMMs-I-0.05, 0.50 J/g for PAN@BMMs-I-0.1, and 0.13 J/g for PAN@BMMs-I-1, along with an increasing AN content in the P(NIPAM-co-AA) matrix. As reported by Gao et al. [44], various lengths of alkyl chains in the AA derivative influenced the balance between hydrophobicity and hydrophilicity in polymer networks.

To verify the characteristic functional groups in the prepared composites, the related samples after vinyl modifying and PAN coating were analyzed using FT-IR spectra. As illustrated in Appendix A, for pure BMMs, the bands at 1080 cm^−1^ and 802 cm^−1^ were both attributed to the stretching vibration of Si–O–Si, and the absorption band at 956 cm^−1^ was ascribed to the vibration features of Si–OH [45]. Comparably, for M-BMMs (Appendix A), the weak bands appeared at 1714 cm^−1^ (v–C=O), 1635 cm^−1^ (v–C=C), 2894, and 2930 cm^−1^ (v–CH_2_); 2985 cm^−1^ (v–CH_3_) could be explained by the vinyl functionalization on the surface of BMMs [46].

Afterward, taking the obtained PAN@BMMs-I-0.1 as an example to compare with either the pure BMMs or M-BMMs, as presented in Appendix A, the additional characteristic bands located at 1645 cm^−1^, 1547 cm^−1^, and 1460 cm^−1^ could be attributed to the C=O, N–H, and C–N stretching vibrations, respectively, which should belong to the amide group [47]. Moreover, the acromion peaks near 1716 cm^−1^ were attributed to the C=O stretching vibrations of the carboxylic acid groups [48]. Noticeably, these regions appeared similar to those of the P(NIPAM-co-AA) (Appendix A), indicating the successful encapsulation of the copolymer onto the vinyl-modified surfaces of the mesoporous BMMs. The typical vibrational absorption peaks of the aromatic ring [49] were observed at 780 cm^−1^ and 760 cm^−1^ for PAN@BMMs-I-0.1 (Appendix A), which aligned well with the infrared spectrum of AN (Appendix A). Subsequently, the I/PAN@BMMs-I-0.1 (Appendix A) exhibited characteristic absorption peaks at 1720 cm^−1^, assigned to the C=O stretching vibration of IBU molecules (Appendix A), which ensured the incorporation of IBU. In PAN@BMMs-I-0.05 (Appendix A), PAN@BMMs-I-1 (Appendix A), PAN@BMMs-II-0.1 (Appendix A), and PAN@BMMs-mix-0.1 (Appendix A), similar results were also observed. The characteristic peaks appeared at 1645, 1547, and 1460 cm^−1^ for amide groups in the PAN shell, 1716 cm^−1^ for the carboxylic acid, and 780 and 760 cm^−1^ for aromatic rings, originating from the NIPAA, AA, and AN components, respectively. These observations further proved that the luminescent groups were doped into the polymer shell of PAN@BMMs.

Figure 2A,B show the SEM and TEM images of representative samples. As depicted in the SEM (Figure 2A (a)) and TEM (Figure 2B (a)) images, M-BMMs exhibited almost-spherical morphologies at a size of 30–50 nm with uniform and worm-like mesopores (around 3 nm), well in agreement with our previous observation of parent BMMs [33]. Noticeably, the cross-linking phenomena obviously appeared in the typical PAN@BMMs-I-0.1 (Figure 2A,B (b)), further verifying that the PAN polymer was coated on the outer surfaces of BMMs. In particular, the corresponding inset images (as shown in Figure 2B (a) and (b)) showed the details of the pore structures of the BMM core and PAN shell features. As can be seen, the mesopore channels of PAN@BMMs-I-0.1 (Figure 2B (b) insert) became indistinct as compared with those of M-BMMs (Figure 2B (a) insert), indirectly confirming that the outer surfaces of the BMM core were covered by a PAN shell.

XPS characterization was used to analyze the elemental compositions of the prepared hybrid composites and their chemical states. As shown in Appendix A, the XPS survey spectra of P(NIPAM-co-AA)@BMMs (Appendix A) and PAN@BMMs-I-0.1 (Appendix A) presented the appearances of O, N, C, and Si elements, showing the surface compositions in the prepared hybrid nanocomposites. In detail, Appendix A showed the N1s spectra of P(NIPAM-co-AA)@BMMs (Appendix A) and PAN@BMMs-I-0.1 (Appendix A), in which the peaks located at 399.7 and 402.1 eV were ascribed to the C–N and N–H bonds [6], respectively. Meanwhile, Appendix A presented their C1s spectra: the peak centered at 284.6 eV was ascribed to the C–C and C–H bonds in AA and NIPAM [50], and the peaks appearing at 286.2 and 287.5 eV were attributed to the C–N bonds in NIPAM and the C=O bonds in AA and NIPAM, respectively. Additionally, the chemical compositions of these samples were further analyzed based on the binding energy regions of C1s, N1s, O1s, and Si in the XPS spectra. The results showed that the atomic ratio of Si was 12.56% in P(NIPAM-co-AA)@BMMs and 26.60% in PAN@BMMs-I-0.1, which was derived from BMMs. Meanwhile, the atomic ratios of C and N were 45.45% and 6.89% in P(NIPAM-co-AA)@BMMs and 14.31% and 1.90% in PAN@BMMs-I-0.1, respectively. Obviously, these C and N elements originated from P(NIPAM-co-AA) or the PAN shell. Based on the abovementioned descriptions, the XPS characterizations clearly indicated that the P(NIPAM-co-AA) copolymer or the PAN shell was successfully coated onto the surfaces of BMM core.

### 3.2. Fluorescence Performances and Fractal Features

To evaluate the aggregation state of the fluorescent AN dispersed in hybrid nanoparticles, its fluorescent property was explored under various conditions. Appendix A exhibited the emission spectra of AN at a concentration of 2 × 10^−3^ mol/L in different solvents. As can be seen, the intense emission of AN in PE, 1,4-Dioxane, EtOH, MeCN, and DMSO was observed at 466 nm (Appendix A), 504 nm (Appendix A), 531 nm (Appendix A), 538 nm (Appendix A), and 535 nm (Appendix A), respectively. Its fluorescence emission wavelength red shifted and the fluorescence intensity decreased rapidly as the solvent polarity increased, as follows: 69 nm for PE, 41 nm for 1,4-Dioxane, 14 nm for EtOH, 7 nm for MeCN, and 10 nm for DMSO. The emission wavelength (*λ_em_* = 466 nm) in PE was the shortest, while the fluorescence intensity in PE was almost 100 times that in DMSO, indicating its fluorescence *λ_em_* was strongly related to the solvents used [49,51]. Therefore, it was speculated that the presence of AN in PE was close to a monomeric state; therefore, PE was selected as a good solvent without background interference in the wavelength range of 465–545 nm for further investigating the fluorophore dispersity.

Figure 3A (a) to (e) showed the fluorescence emission spectrum of AN in PE solvent at various concentrations, and the variations of the emission wavelength was plotted in the inset image. As can be seen, the maximum emission wavelength slightly increased from 465 to 468 nm with the concentration ranging from 1 × 10^−5^ to 1 × 10^−2^ mol/L, but the solid AN (Figure 3A (f)) exhibited an emitting fluorescence at 545 nm. These observations suggested that the blue emission at 465 nm and the other emission at 545 nm were associated with the monodisperse and aggregated states of AN. However, the mechanism of the solvent-dependent fluorescence emission profiles (as shown in Appendix A) was different from that of the concentration-dependent fluorescence emission profiles (as shown in Figure 3A) in the monomeric and aggregated states of the used AN.

Experimentally, we tentatively assigned the state of fluorescent AN by the emission wavelength and three different degrees of dispersion exhibited: full aggregation (545 nm), intermediate (465 nm to 545 nm), and completely dispersion (465 nm), which was similar to the method that reported by Wang et al. [52], where the macro-dispersion of inorganic fillers in organic–inorganic composites was preliminary discussed by using fluorescent imaging.

Figure 3B presents the fluorescence spectra of the synthesized PAN@BMMs. It can be seen in Figure 3B (a) and (b) that no emission appeared over the range of 450–650 nm in the fluorescent spectra for the obtained M-BMMs and P(NIPAM-co-AA). In contrast, the intense fluorescent emissions of PAN@BMMs-I-0.1, PAN@BMMs-II-0.1, and PAN@BMMs-mix-0.1 were detected at 476 nm (Figure 3B (d)), 499 nm (Figure 3B (f)), and 508 nm (Figure 3B (g)). As compared with 545 nm of AN solid (Figure 3A (f)), PAN@BMMs-I-0.1 prepared by a “one-pot” method had significantly blue shifted (69 nm), indicating that the AN distributed on the surfaces was close to the homogeneous monomer state, and no visible tendency to form aggregates was observed. Meanwhile, the maximum emission wavelength (508 nm) of the physically mixed PAN@BMMs-mix-0.1 (Figure 3B (g)) was closer to the 545 nm of the AN-aggregate solid (Figure 3A (f)). Notably, the decrease in the fluorescent intensity of PAN@BMMs-mix-0.1, as compared to that of the PAN@BMMs-I-0.1, should be related to the washing out of the physically adsorbed AN in the process of removing CTAB. Consequently, the stability of AN binding to PAN@BMMs during the swelling or releasing process was also discussed.

Obviously, the dispersity of AN on the PAN networks obtained via a one-pot approach was better than that obtained via the two-step or physically mixed methods; therefore, the one-pot method was selected to further demonstrate the effects of the dispersed AN amount on its fluorescent performance. Figure 3B (c) to (e) shows the fluorescent spectra of PAN@BMMs doped with an AN amount of 0.05, 0.1, and 1% via the one-pot method. As can be seen, the maximum emission wavelength was red-shifted from 471 nm for PAN@BMMs-I-0.05 (Figure 3B (c)) to 476 nm for PAN@BMMs-I-0.1 (Figure 3B (d)), which was accompanied by a further redshift to 488 nm for PAN@BMMs-I-1 (Figure 3B (e)) when the doped-AN amount increased up to 1.0%, which should be assigned to the intermediate dispersions of AN according to the abovementioned aggregative distribution of AN. Compared with the AN aggregated state (545 nm, as shown in Figure 3A (f)), the blue shift (74 nm for PAN@BMMs-I-0.05, as shown in Figure 3B (c); 69 nm for PAN@BMMs-I-0.1, as shown in Figure 3B (d); and 57 nm for PAN@BMMs-I-1, as shown in Figure 3B (e)) seemed to be weaker with the increase of the NA-doping amount, meaning that the lower content of the doped AN was conductive to transformations from the aggregation state into the monodispersed state. As expected, the increase in the AN-doped amount from 0.05 to 1% should be useful in promoting the fluorescent intensity of PAN@BMMs. In addition, these spatial dispersion states of AN doped in PAN@BMMs were further depicted via the fractal demonstrations.

TGA profiles were applied to illustrate the doping of AN in the network of polymers. As shown in Appendix A, the weight loss at 30–200 °C in all related PAN@BMMs was associated with the removal of the physically adsorbed and chemically adsorbed water inside the mesoporous channels [53]. For M-BMMs (Appendix A), the weight loss of about 5.7 wt% at the range of 200–850 °C was mainly attributed to the decomposition of organic functional groups, while that of pure BMMs was 3.2 wt% (Appendix A); consequently, the calculated MPS-modified content was 2.5 wt%.

Thereafter, the weight loss of the AN-doped samples was observed in the temperature range of 200–850 °C, showing 36.6 wt% for PAN@BMMs-I-0.05 (Appendix A), 38.2 wt% for PAN@BMMs-I-0.1 (Appendix A), 39.6 wt% for PAN@BMMs-I-1 (Appendix A), 37.3 wt% for PAN@BMMs-II-0.1 (Appendix A), and 34.7 wt% for PAN@BMMs-mix-0.1 (Appendix A). As presented in Appendix A, the P(NIPAM-co-AA) and AN were almost completely pyrolyzed at 500 °C and underwent about 96.9 wt% and 99.2 wt% weight loss. These phenomena further proved the successful encapsulation of P(NIPAM-co-AA) doped with different AN aggregations. The weight loss should be proportional to the amount of the coated PAN. However, when the feeding amounts (0.05–1%) of the AN doped in PAN@BMMs were 0.05–1%, the weight loss remained almost constant, showing 38.4 wt% for PAN@BMMs-I-0.05 (Appendix A) and 38.6 wt% for PAN@BMMs-I-1 (Appendix A). These results indicated that the effect of the AN doped in PAN@BMMs on the PAN shell attached to the surfaces of the BMMs core was unobvious.

Furthermore, the fluorescence decay lifetimes were measured to elucidate the underlying luminous mechanism of the fluorophores. Figure 3C showed the time-resolved fluorescence decay curves of prepared PAN@BMMs samples; the two-exponential fitting lifetime results are listed in Appendix A. The results presented that the AN (Figure 3C (h)) displayed a two-lifetime (*τ*_1_ = 3.78 ns, *τ*_2_ = 12.27 ns). Similar results were also demonstrated by Duhamel et al. [54]; the probable reason for this was the presence of the monomers (~3.3 ns) and excimers (~10 ns) in doped AN. In particular, the appearances of the excimers were generated by conjugations between the excited naphthalene ring and the ground-state naphthalene ring [54]. These phenomena were well elucidated above by the fluorescence spectra of AN at various concentrations in PE (Figure 3A), in which the fluorescence emission wavelength of solid AN (545 nm) was clearly red shifted with respect to that of the AN monomer (465 nm). Accordingly, the longer lifetime (*τ*_2_ = 12.27 ns) presented as the formed AN excimer; the other was attributed to the AN monomers with a short lifetime of 3.78 ns. The pre-exponential factors reflected the amount of AN monomers relative to the excimers.

Comparably, the fluorescence decays of hybrid PAN@BMMs also presented double decay lifetimes, showing 3.59 ns and 10.62 ns for PAN@BMMs-I-0.1 (Figure 3C (d)), 2.64 ns and 8.56 ns for PAN@BMMs-II-0.1 (Figure 3C (f)), and 3.53 ns and 9.00 ns for PAN@BMMs-mix-0.1 (Figure 3C (g)). Obviously, both of the lifetimes of doped-AN PAN@BMMs were shorter than those of AN (Figure 3C (h)), indicating the occurrence of quenching. On the basis of experiments by Hwang et al. [55], the variations in the long lifetime of the excimers might be a consequence of the electron-absorbing groups (–COOH and –CONH–) in the PAN copolymers [55]. The pre-exponential factors of the excimers (*B*_2_) prepared using the two-step (*B*_2_ = 59.96) and mixing methods (*B*_2_ = 75.88) were higher than those obtained by the one-pot process (*B*_2_ = 59.48). These results suggest that the AN molecules were more likely to agglomerate in the PAN@BMMs prepared using two-step and mixing methods, thus promoting the formation of excimers, but more evenly dispersed in the PAN@BMMs prepared by the one-pot process.

Meanwhile, the lifetimes of PAN@BMMs-I-n gradually prolonged with the increase of the doped AN, showing 3.35 ns and 10.50 ns for PN A@BMMs-I-0.05 (Figure 3C (c)), 3.59 ns and 10.62 ns for PAN@BMMs-I-0.1 (Figure 3C (d)), and 3.89 ns and 10.92 ns for PAN@BMMs-I-1 (Figure 3C (e)). The pre-exponential factor (*B*_2_) of AN excimers was increased from 60.23 for PNA@BMMs-I-0.05 to 63.83 for PAN@BMMs-I-1, verifying that the serious aggregation of the doped AN led to the appearance of a large number of excimers. Therefore, these fluorescence lifetime results further showed that the fluorescence performances were closely associated with their various dispersion behaviors.

As illustrated in Figure 3D, the absorption spectrum of P(NIPAM-co-AA)@BMMs (Figure 3D (a)) presented partial overlapping with the emission band of AN (Figure 3D (b)), suggesting resonance energy transfer from AN to P(NIPAM-co-AA)@BMMs. The excited AN is conductive to provoke an electron donor–acceptor interaction between the unbound electron pair of the nitrogen atom at the C-4 position toward the electron-accepting 1,8-dicarboxylic groups [24]. This intramolecular electron charge transfer easily results in a decrease in the fluorescence lifetime of the hybrid composites.

In order to elucidate the microscopic dispersion state of AN in PAN@BMMs, the fractal analysis theory was introduced based on SAXS patterns. Figure 4A displays the fractal characteristics of all BMM-based samples; corresponding PDDF curves are presented in Figure 4B. As can be seen in Figure 4A, the slope of each curve (yellow straight line) was obtained with a linear fit using the least square method in low-*q* regions (region), and thus the following fractal dimension (*d_m_* or *d_s_*) values were determined by power-law decay. We noticed that the slope values of all related samples were between two and three in the low-*q* region (−2.20 < *lnq* < −1.52); their behaviors possessed the *d_m_* characteristics, corresponding their *d_m_* values ranging between two and three, implying that the PAN@BMMs were indicative of loose and porous structures.

As shown in Figure 4A (a) and (b), the *d_m_* values slightly increased after vinyl modification from 2.06 (±0.04) for pure BMMs (Figure 4A (a)) to 2.14 (±0.03) for M-BMMs (Figure 4A (b)), indicating that the structure of the M-BMMs tended to be compact, which was responsible for the formation of the Si-O-Si structure. These results were also evident from the pore size distributions (Appendix A). Additionally, the fractal structure of polymer-coated P(NIPAM-co-AA)@BMMs (Figure 4A (c)) was more compact than that of M-BMMs, according to larger *d_m_* value (2.39 (±0.04)) of P(NIPAM-co-AA)@BMMs than that of M-BMMs (2.14 (±0.04)). Therefore, these results indicated that the P(NIPAM-co-AA) was successfully grafted onto the surface of M-BMMs.

The *d_m_* value of the obtained PAN@BMMs-I-0.1 (Figure 4A (e)) was about 2.53 (±0.04), which was smaller than the *d_m_* values of 2.61 (±0.04) for PAN@BMMs-II-0.1 (Figure 4A (g)) and 2.82 (±0.04) for PAN@BMMs-mix-0.1 (Figure 4A (h)). These variations of fractal features may be related to different dispersion states of fluorescent molecules that vary by doping method. Meanwhile, the *d_max_* value slightly increased from 2.48 (±0.04) for PAN@BMMs-I-0.05 (Figure 4A (d)) to 2.70 for PAN@BMMs-I-1 (Figure 4A (f)), indicating that the introducing of a larger AN cluster onto the PAN shell in PAN@BMMs tended towards the densification evolution.

The PDDF profiles revealed the largest particle size (*d_max_*) via the intersection between the curves and the horizontal axis [56]. As shown in Figure 4B, the *d_max_* value was about 41.7 nm for pure BMMs (Figure 4B (a)) and 44.7 nm for M-BMMs (Figure 4B (b)), experiencing little change during the vinyl modification. However, the detectable differences were observed after PAN coating: the *d_max_* value was increased to 67.5 nm for P(NIPAM-co-AA)@BMMs (Figure 4B (c)), 70.3 nm for PAN@BMMs-I-0.05 (Figure 4B (d)), 72.6 nm for PAN@BMMs-I-0.1 (Figure 4B (e)), 93.1 nm for PAN@BMMs-I-1 (Figure 4B (f)), 74.7 nm for PAN@BMMs-II-0.1 (Figure 4B (g)), and 88.9 nm for PAN@BMMs-mix-0.1 (Figure 4B (h)), seeming to be a steadily increasing tendency.

Additionally, most BMM-based samples displayed a bell-shaped profile, implying the appearances of the ellipsoid-shape nanoparticles [57]. However, it is worth noting that the shape of the physical-mixing sample (PAN@BMMs-mix-0.1) was significantly different from those of the one-pot and two-step process samples. Figure 4B (h) exhibited bimodal distribution, probably indicating the presence of two substances. In this regard, we could speculate that the overwhelming majority of AN molecules were successfully doped into the inner-polymer chain prepared by the one-pot or two-step methods rather than mixed onto the outer-surface of the BMMs. Consequently, the PAN@BMMs-mix-0.1 was used as a reference sample for the following investigation of fluorescence stability and AN-dispersion behaviors.

### 3.3. The Dispersion State of AN in PAN@BMMs during Swelling–Shrinking Process

Considering the influences of the temperature- and pH-responsive performance on the monodisperse of fluorescent AN during the swelling–shrinking process, the sensitivity of the resultant PAN@BMMs was systematically investigated using the *D_h_* values measured using the DLS method in PBS at pH values of 2.0 and 7.4 at 25 °C and 37 °C. In particular, the particle size distribution of pure BMMs (Appendix A) showed that the average *D_h_* value was around 196 nm. In addition, the variation of mean *D_h_* values with different swelling and shrinking times was obtained in Figure 5A,B. As can be seen, the mean *D_h_* value of PAN@BMMs-I-n increased with the prolonging of swelling time under pH 7.4 at 25 °C (Figure 5A) but decreased abruptly in pH 2.0 at 37 °C (Figure 5B). For example, the *D_h_* value of PAN@BMMs-I-0.1 was 490 nm for 1 h, 815 nm for 3 h, 931 nm for 8 h, 963 nm for 12 h, and eventually 1668 nm for 24 h at pH 7.4 at 25 °C (Figure 5A (b)) but 931 nm for 1 h, 808 nm for 3 h, 778 nm for 8 h, 754 nm for 12 h, and 658 nm for 24 h at pH 2.0 at 37 °C (Figure 5B (b)). Similar trends were also present in PAN@BMMs-I-0.05 (Figure 5A (a) and B (a)), PAN@BMMs-I-1 (Figure 5AB (c)), PAN@BMMs-II-1 (Figure 5AB (d)), and PAN@BMMs-mix-0.1 (Figure 5AB (e)). The *D_h_* values of the swollen PAN@BMMs in pH 7.4 at 25 °C were up to 1377 nm for PAN@BMMs-I-0.05 (Figure 5A (a)), 2001 nm for PAN@BMMs-I-1 (Figure 5A (c)), 1819 nm for PAN@BMMs-II-0.1 (Figure 5A (d)), and 1304 nm for PAN@BMMs-mix-0.1 (Figure 5A (e)), while their shrunken *D_h_* values in pH 2.0 at 37 °C were 759 nm (Figure 5B (a)), 536 nm (Figure 5B (c)), 613 nm (Figure 5B (d)), and 526 nm (Figure 5B (e)). In essence, the swelling behavior was due to the electrostatic repulsive forces among –COO^–^ originating from the deprotonation of the –COOH groups in PAN chains when the pH was above the *pKa* of PAA (4.8), whereas below it, the intra-molecular hydrogen bonding between the polymer crosslinking networks was formed due to the protonated –COOH groups [58], resulting in the shrinking of the P(NIPAM-co-AA) at pH 2.0. This swelling–shrinking feature was one of the hallmark characteristics of the lesion microenvironment, endowing PAN@BMMs as nanocarriers with great potential in controlled release for drug delivery.

Meanwhile, the fluorescent emission spectra of the swollen PAN@BMMs (S) and the swelling filtered solution (L) over time were obtained, which is important for fluorescent stability and intensity during the drug administration or swelling of the drug carrier.

As illustrated in Appendix A, with the prolonged swelling time in pH 7.4 at 25 °C, a slightly decreased intensity at 488 nm was observed in the fluorescent spectrum of the PAN@BMMs-I-0.1 (S) but this was much stronger than that of the filtrate (L) at 465 nm. Interestingly, a slight blue shift from 488 to 485 nm appeared in the emission spectrum of the filtrate (L) compared with that of the solid samples. On the basis of the AN-monodisperse experiment results (as shown in Figure 3A), these observations implied that the doped AN almost existed as a multimer in PAN@BMMs-I-0.1 during the swelling process, whereas solely small amounts of AN in the filtrate presented close to the monomeric state (465 nm).

Similarly, the fluorescence intensity at 483 nm for PAN@BMMs-I-0.05 (Appendix A), at 495 nm for PAN@BMMs-I-1 (Appendix A), and at 505 nm for PAN@BMMs-II-0.1 (Appendix A) also slightly decreased in pH 7.4 at 25 °C, accompanied by an unobvious intensity increase at 485 nm for the solution. Notably, the fluorescence intensity at 508 nm for the physically mixed PAN@BMMs-mix-0.1 (Appendix A) decreased, and there was a pronounced redshift to 542 nm in the filtrate (L), being closer to that of the AN-aggregated state (at 545 nm, as shown in Figure 3A). The most probably reason for this is that the physically adsorbed AN in PAN@BMMs-mix-0.1 is dissolved into the filtrated solution.

However, at pH 2.0 and 37 °C, the fluorescence intensity at 490 nm for PAN@BMMs-I-0.1 (S) was significantly reduced (Appendix A), and the gradual increase in fluorescence intensity at 486 nm for filtrates (L) was more evident than that of the swollen samples at pH 7.4 and 25 °C (Appendix A). This was probably due to the swelling of the PAN shell occurring under weak bases, leading to its network opening and causing the diffusion of AN into the solvent. When the acidic conditions and higher temperatures caused the dual pH- and temperature-responsive PAN shell to shrink, more fluorescent NA diffused to the solution, and the wavelength of the PAN@BMMs-I-0.1 (L) red shifted from 486 nm for 1 h towards 488 nm for 24 h (Appendix A).

Similarly, the solid PAN@BMMs-I-0.05 (Appendix A), PAN@BMMs-I-1 (Appendix A), and PAN@BMMs-II-0.1 (Appendix A) presented stable fluorescence intensity at 483, 497, and 507 nm, accompanied by blue-shifted emissions of their filtrate at 481, 488, and 486 nm. In comparison, these blueshift phenomena cannot be detected over the PAN@BMMs-mix-0.1; the wavelengths of its filtrate were redshifted towards 524 nm (Appendix A), verifying occurrences of the aggregated clusters of NA in the filtrated solvent. This observation clearly interpreted the weak physical adsorption between AN and P(NIPAM-co-AA) in PAN@BMMs-mix-0.1. Therefore, we could speculate that the AN doped in PAN@BMMs were actually incorporated into the networks of the PAN shell through intermolecular hydrogen bonding between the –NH_2_ of AN and the –CONH–, –COOH of P(NIPAM-co-AA).

More specifically, the corresponding microstructure evolutions for monitoring the uniformity of AN dispersion have been successfully evaluated based on the SAXS patterns. The fractal dimension and the *d_max_* values of the related samples in the swelling (pH 7.4 at 25 °C) and shrinking (pH 2.0 at 37 °C) durations are presented in Figure 6.

The results showed that all samples retained the typical *d_m_* features during the swelling or shrinking processes, implying that the networks of PAN@BMMs were indicative of loose and porous structures [59]. In detail, we observed that the *D_m_* values of PAN@BMMs-I-0.1 (Figure 6A) significantly decreased from 2.74 (±0.04) for 1 h to 2.52 (±0.04) for 3 h, 2.47 (±0.04) for 8 h, 2.39 (±0.04) for 12 h, and 2.24 (±0.04) for 24 h at pH 7.4 and 25 °C, indicating that the coated-PAN shell in PAN@BMMs-I-0.1 gradually evolved into a loose structure during the swelling process. Similar results were also present in other samples: the *d_m_* values of PAN@BMMs-I-0.05 (Appendix A), PAN@BMMs-I-1 (Appendix A), PAN@BMMs-II-0.1 (Figure 6C), and PAN@BMMs-mix-0.1 (Figure 6E) showed the reduced tendencies of 2.53–2.20, 2.81–2.46, 2.85–2.27, and 2.87–2.38, respectively. When their shrinking was conducted at pH 2.0 and 37 °C, the *D_m_* value of PAN@BMMs-I-0.1 (Figure 6B) gradually increased from 2.36 (±0.04) for the initial 1 h to 2.48 (±0.04) for 3 h, 2.53 (±0.04) for 8 h, 2.58 (±0.04) for 12 h, and then achieved 2.63 (±0.04) until 24 h, showing a progressively densified microstructure. The same phenomenon of an increased tendency was also found in the PAN@BMMs-I-0.05 (Appendix A), PAN@BMMs-I-1 (Appendix A), PAN@BMMs-II-0.1 (Figure 6D), and PAN@BMMs-mix-0.1 (Figure 6F), with *d_m_* values of 2.30–2.59, 2.53–2.80, 2.48–2.71, and 2.43–2.78. Obviously, these results were attributed to the swelling (or shrinking) behaviors of the PAN shell under a weakly alkaline (acid) medium of pH 7.4 (or 2.0) at 25 (or 37) °C [60].

Additionally, the swelling–shrinking process of the PAN shell was accompanied by a dramatic *d_max_* values change. As can be seen in Figure 6 (insert), the PDDF curves of the swollen or shrunken PAN@BMMs displayed a bell-shaped profile, implying the appearances of the ellipsoid-shaped morphologies. As shown in Figure 6A (insert), the *d_max_* values of PAN@BMMs-I-0.1 at pH 7.4 and 25 °C were about 76.4 nm after swelling for 1 h, 77.1 nm for 3 h, 87.4 nm for 8 h, 88.9 nm for 12 h, and even up to 91.3 nm for 24 h, respectively. Obviously, a significant shift of the particle to a larger size occurred with an increase in swelling time. In addition, under pH 2.0 at 37 °C (Figure 6B (insert)), its *d_max_* values decreased to 80.9 nm for 1 h, 76.5 nm for 3 h, 71.7 nm for 8 h, 67.9 nm for 12 h, and 65.4 nm for 24 h.

As shown in Appendix A, similar results were also obtained in PAN@BMMs-I-0.05 (Appendix A), PAN@BMMs-I-1 (Appendix A), and PAN@BMMs-II-0.1 (Figure 6C,D (insert)) in which, in pH 7.4 at 25 °C, their *d_max_* values were 49.8 to 80.8 nm for PAN@BMMs-I-0.05, 51.7 to 84.1 nm for PAN@BMMs-I-1, and 42.6 to 78.7 nm for PAN@BMMs-II-0.1. Comparatively, after immersion in an acidic (pH 2.0) environment at 37 °C, their *d_max_* values decreased from 66.8 nm to 33.4 nm for PAN@BMMs-I-0.05, from 68.6 nm to 59.3 nm for PAN@BMMs-I-1, and from 65.3 nm to 51.3 nm for PAN@BMMs-II-0.1. Whereas, as illustrated in the PDDF profiles of the reference PAN@BMMs-mix-0.1, over swollen or shrunken time it exhibited unsymmetrical curves, representing the microstructure inhomogeneity that was possibly caused by the formation of the doped AN aggregates. The *d_max_* value varied from 74.6 to 90.1 nm in pH 7.4 at 25 °C (Figure 6E (insert)) and then conversely from 89.5 to 75.6 nm in pH 2.0 at 37 °C (Figure 6F (insert)). These results were consistent with those of the DLS analysis, evidently verifying the pH- and temperature-responsive behaviors of PAN@BMMs, which could be conducive to controlled drug releasing.

### 3.4. Fluorescent Properties and Fractal Evolution during Drug Loading and Release

The HPLC measurements indicated that the PAN@BMMs-I-0.05, -0.1, -1, PAN@BMMs-II-0.1, and PAN@BMMs-mix-0.1 can be IBU loaded with the LC% of 12.8, 11.3, 8.3, 15.9, and 12.9%, respectively, suggesting that the IBU can be embedded into the cores of BMMs in PAN@BMMs. Subsequently, the temperature- and pH-responsive releases of IBU from I/PAN@BMMs were investigated in the simulated PBS fluids at pH 7.4 25 °C and pH 2.0 at 37 °C. As shown in Figure 7A, only about 17.3, 19.8, 21.6, 23.9, and 26.0% of IBU was released from I/PAN@BMMs-I-0.05 (Figure 7A (a)), I/PAN@BMMs-I-0.1 (Figure 7A (b)), I/PAN@BMMs-I-1 (Figure 7A (c)), I/PAN@BMMs-II-0.1 (Figure 7A (d)), and I/PAN@BMMs-mix-0.1 (Figure 7A (e)) over 24 h under the conditions of pH 2.0 at 37 °C. Notably, almost no initial burst release was observed. By contrast, a fast and large amount of IBU was released in pH 7.4 at 25 °C (Figure 7B); the cumulative release percent reached about 67.4% for I/PAN@BMMs-I-0.05 (Figure 7B (a)), 72.1% for I/PAN@BMMs-I-0.1 (Figure 7B (b)), 79.2% for I/PAN@BMMs-I-1 (Figure 7B (c)), 83.2% for I/PAN@BMMs-II-0.1 (Figure 7B (d)), and 88.1% for I/PAN@BMMs-mix-0.1 (Figure 7B (e)) after 24 h.

At pH 7.4 at 25 °C, the swelling behavior of the PAN shell in PAN@BMMs was conductive to opening the pores, resulting from the electrostatic repulsion of the deprotonated carboxyl group in PAN [61], and thereby leading to the ease of diffusion of the loaded drug from the opened pore channels and a higher release amount. In contrast, in pH 2.0 at 37 °C, the PAN shell was shrunken due to the formation of hydrogen binding among carboxyl groups and the hydrophobicity among the hydrophobic groups (–CH(CH_3_)_2_) at the temperature above the LCST of PNIPAM [62]. These similar observations were confirmed by the fact that the hydrodynamic diameter of the PAN@BMMs in pH 7.4 at 25 °C was larger than that in pH 2.0 at 37 °C. Consequently, the shrinking of the PAN shell in PAN@BMMs could be modulated in acidic PBS (pH = 2.0) at 37 °C, which effectively encapsulated the mesopores of BMM core, preventing IBU from diffusing through the mesoporous channels.

Thereafter, the photoluminescence spectra of the IBU-released samples were also investigated at pH 7.4 at 25 °C and pH 2.0 at 37 °C. As shown in Appendix A, the fluorescence intensity at 485 nm for I/PAN@BMMs-I-0.1 (S) slightly decreased with the extension of the release time at pH 7.4 at 25 °C, but it was stronger than that of the filtrated solutions (L) located at 482 nm. Interestingly, the peak intensity of all filtrated solutions was shifted to a shorter wavelength region compared with that of the I/PAN@BMMs-I-0.1 solid, suggesting that the doped AN existed as a monomer state in the filtrates. Meanwhile, in pH 2.0 at 37 °C (Appendix A), the fluorescence intensity at 486 nm for solid I/PAN@BMMs-I-0.1 (S) decreased relatively to the prolonged release time; in contrast, the emission peak of filtrates (L) was enhanced and further redshifted to 492 nm. The reason for this may be related to the higher extrusion of doped AN from the shrunken PAN networks. Fortunately, the overall PL intensity of filtrates (L) was much lower than that of I/PAN@BMMs-I-0.1 (S). Similar results were also observed in other I/PAN@BMMs, as shown in Appendix A. These results indicated that the PAN@BMMs maintain a good fluorescent stability during the IBU release process, which is especially beneficial for monitoring or tracking the drug delivery to function as a luminescent probe.

The fractal evolutions of the pore microstructure in PAN@BMMs occurred during the IBU loading and controlled release from PAN@BMMs carriers. Taking PAN@BMMs-I-0.1 as an example, its fractal features and PDDF curves were both introduced to further describe the IBU loading and releasing performances. According to the abovementioned low-*q* region of swollen PAN@BMMs (−2.20 < *lnq* < −1.52), all samples presented a typical fractal feature with a good correlation *R^2^* (> 0.99).

Figure 8A presented the time-dependent fluorescent spectra of the solid sample and the corresponding filtered solution. The peak (475 nm) intensity of the solution (L) gradually increased over loading time up to 48 h (Figure 8A-L), whereas a slightly decreased intensity of the peak (465 nm) was observed for I/PAN@BMMs-I-0.1 solid (Figure 8A) due to NA leaching from the PAN shell. Additionally, Figure 8A (insert) presented the IBU-loading capacity of PAN@BMMs-I-0.1, showing 7.9% for 1 h, 9.6% for 5 h, 10.4% for 12 h, and up to 11.3% for 48 h.

As shown in Figure 8B, the *d_m_* value of PAN@BMMs-I-0.1 increased from 2.88 (±0.02) for 1 h (Figure 8B (a)) to 2.93 (±0.02) for 5 h (Figure 8B (b)). Interestingly, when the IBU-loading amount increased to 11.3%, its fractal features were transformed from *d_m_* to *d_s_*, the corresponding *d_s_* values were 2.93 (±0.02) for 12 h (Figure 8B (c)), 2.55 (±0.04) for 24 h (Figure 8B (d)), and 2.49 (±0.04) for 48 h (Figure 8B (e)), implying variations from loose to dense structures with a rough surface along with a decrease in their mean mesopore size and volume (as shown in Appendix A). These observations indicated that IBU was indeed loaded into the mesoporous channels of PAN@BMMs-I-0.1, in agreement with Varga’s report [63]. Additionally, as the drug-loading time increased, the PDDF curves (Figure 8B (insert)) showed that the *d_max_* value of the particle increased from 51.3 nm at 1 h to 58.0 nm at 5 h, 61.1 nm at 12 h, 75.4 nm at 24 h, and 76.5 h at 48 h, suggesting the successful encapsulation of IBU to the BMM core.

However, the fractal features of I/PAN@BMMs-I-0.1 were conversely transformed from *d_s_* to *d_m_* during IBU releasing. As can be seen in Figure 8C, the *d_s_* (the value of 2.68 (±0.04) for 1 h (Figure 8C (a)) and 2.92 (±0.04) for 3 h (Figure 8C (b))) reconverted to *d_m_*, and then the *d_m_* value decreased to 2.80 (±0.04) at 8 h (Figure 8C (c)) and 2.75 (±0.04) at 12h (Figure 8C (d)) during the prolonged IBU releasing and reached 2.69 (±0.04) at 24 h (Figure 8C (e)) when the releasing percent was 19.8% at equilibrium time under pH 2.0 at 37 °C, gradually showing the loose microstructures. The corresponding *d_max_* value (Figure 8C (insert)) of the I/PAN@BMMs-I-0.1 particles decreased from 77.6 nm for 1 h to 64.2 nm for 3 h, 58.6 nm for 8 h, 49.7 nm for 12 h, and 39.0 nm for 24 h. One of the main reasons for this was the shrinking of the soft PAN shell, rather than variations in the static rigid BMM core. These results suggested that the drug was released from the mesopore channels of the BMM core.

Identically, with the drug released in pH 7.4 at 25 °C, Figure 8D also showed that the *d_s_* (value of 2.81 (±0.05) for 1 h (Figure 8D (a))) shifted to *d_m_* (value of 2.92 (±0.04) for 3 h (Figure 8D (b))), and the *d_m_* value decreased to 2.80 (±0.04) for 8 h (Figure 8D (c)) and 2.72 (±0.04) for 12 h (Figure 8D (d)). However, the *d_m_* value reached 2.59 (±0.04) at 24 h (Figure 8D (e)) when the IBU-releasing percent was 72.1% at equilibrium time under pH 7.4 at 25 °C, lower than that (2.69 (±0.04)) in pH 2.0 at 37 °C (Figure 8C), which may be owing to the significant amounts of IBU released from the mesoporous channels.

Additionally, the *d_max_* values (Figure 8D (insert)) of the drug-released I/PAN@BMMs-I-0.1 were increased from 59.9 nm at 1 h to 65.7 nm at 3 h, 66.1 nm at 8 h, 77.7 nm at 12 h, and 78.2 nm at 24 h, which are higher than those in pH 2.0 at 37 °C. The possible reason for this was that the drug diffusing from the mesoporous channels of the BMM core resulted in the uptake of water during the swelling of the PAN shell in pH 7.4 at 25 °C.

### 3.5. Cytotoxicity Study

Considering that the biocompatibility of an ideal drug carrier should be imaged via in vitro experiments [64,65], the cytotoxicity of synthetic PAN@BMMs was evaluated in HeLa cells. The CCK-8 assay results are shown in Figure 9. The cell viability of BMMs and copolymer P(NIPAM-co-AA) remained at 89.8% (Figure 9 (a)) and 87% (Figure 9 (b)) at concentrations as high as 100 μg/mL for BMMs and 50 μg/mL for P(NIPAM-co-AA), revealing a low cytotoxicity, which is consistent with the previous reports of others [66,67].

Although the toxicity of the doped AN on HeLa cells was somewhat dependent on the dose, as can be seen in Figure 9 (f), the viability was more than 95% at 1 μg/mL. However, the AN-doped PAN@BMMs exhibited much less cytotoxicity, showing that the cell viabilities of the PAN@BMMs-I-0.05, -0.1, and -1 samples were more than 85% at a concentration of 100 μg/mL (Figure 9 (c)), 50 μg/mL (Figure 9 (d)), and 20 μg/mL (Figure 9 (e)) after 48 h culture. The possible reason for this was the fact that the used AN was effectively wrapped inside the cross-linked networks of PAN copolymers. Therefore, these cytotoxicity results demonstrated that PAN@BMMs have a low cytotoxicity and safety as drug carriers, showing the potential in vivo applications.

### 3.6. Cellular Uptake, Localization, and Imaging

The cellular uptake behavior of PAN@BMMs is essential for the target drugs to realize their therapeutic effects. Thus, the time-dependent internalization of PAN@BMMs-I-0.1 was observed using a CLSM. Benefiting from very low cytotoxicity, the in vitro cellular uptake behavior of the PAN@BMMs-I-0.1 was carefully evaluated at a concentration of 20 μg mL^−1^.

As shown in Figure 10, the clear green fluorescence that represented the fluorescent PAN@BMMs-I-0.1 was observed in the cytoplasm of the cell, indicating that their metabolic pathway can be monitored with fluorescence imaging. After cell culturing with PAN@BMMs-I-0.1 for 6 h, a small amount of fluorescence was seen near cell membranes, and the fluorescence intensity of the cytoplasm increased slightly compared with the control sample in the same time interval (Appendix A). Remarkably, brighter green fluorescent nanoparticles localized within the cytoplasm after 12 h of incubation implied successful taking up by the cells. After 24 h, a large number of PAN@BMMs-I-0.1 were extensively distributed in the cytoplasmic area of most cells. Moreover, a dramatic enhancement in the cytoplasm’s fluorescent intensity was observed compared with after 24 h for the control sample (Appendix A). These results demonstrated that PAN@BMMs-I-0.1 as a drug carrier was internalized in a time-dependent manner and gradually accumulated at the cell, which was related to the appearances of the carboxyl groups (–COOH) existing in the PAN-coated shell [68]. In addition, the green fluorescent intensity of PAN@BMMs was strong and stable, probably due to the presence of the AN being wrapped in the cross-linked networks of PAN copolymers, showing the potential application for intravital imaging.

## 4. Conclusions

The fluorescent hybrid PAN@BMMs based on core–shell structures were successfully constructed, in which three binding methods (one-pot, two-step, and mixing methods) and three composite matrixes (AN-doped amount of 0.05%, 0.1%, and 1%) were demonstrated via various characterizations. The fractal features suggested that the doped AN were uniformly dispersed in PAN@BMMs. Meanwhile, the possible principal associations between the fractal structures of the micro-dispersed fluorophore and their fluorescent performances were elucidated, and the PL spectra of PAN@BMMs revealed a red-shifted emission wavelength (471–508 nm) as the AN-doped amount increased. In particular, the *d_m_* evolution in values of 2.75–2.25 and *d_max_* values of 76.4–91.3 nm were present along with the extension of the swelling time for PAN@BMMs-I-0.1. The transformation between *d_s_* and *d_m_* further proved the microstructural alterations during the IBU-release process. Additionally, the long fluorescence decay two-lifetimes (3.59 and 10.62 ns) of PAN@BMMs-I-0.1 were assessed, evaluating its charge transfer behavior of the chromophore. Finally, the in vitro cell assays of PAN@BMMs on Hela cells showed extremely low cytotoxicity at low concentrations (50 µg/mL), and CLSM observation indicated that PAN@BMMs-I-0.1 with the green fluorescence could be internalized in a time-dependent manner. Therefore, thermal-/pH-sensitive PAN@BMMs could be used as tracers for various applications in controlled drug delivery and assay bioimaging.

## Data Availability

The data presented in this study are available on request from the corresponding author.

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
