# Peer review of "Dispersion Performances of Naphthalimides Doped in Dual Temperature- and pH-Sensitive Poly (N-Isopropylacrylamide-co-acrylic Acid) Shell Assembled with Vinyl-Modified Mesoporous SiO2 Core for Fluorescence Cell Imaging"

_polymers, 2023, doi:10.3390/polym15102339_

Round 1
Reviewer 1 Report
Nothing is characterized in the study presented by Xu et al. Whether the copolymer of isopropyl acrylamide and acrylic acid is actually bound (or not) to the surface of the bimodal mesoporous silica particles and whether the dye (AN) is covalently (or not) bound to the particles is not known. Consequently, no information is provided about the content of polymer and dye bound to the particles, which means that nothing is known about the particles.
The manuscript is poorly written, uses too many abbreviations that ensure that the reader cannot follow the rare details provided about the study and it contains too many inaccurate statements.
I had started a list of detailed points that should be clarified but the study is so fundamentally wrong that it should be started from scratch.
Such a study should never have been allowed to proceed to the reviewing stage.
Author Response
Responses to the comments of Reviewer 1#

Reviewer 2 Report
General comments:
The manuscript may have some interesting points and correlations but it is very difficult to follow. There are no sections or data tables to provide a concise summary of the results. The molecular insight may be strengthened by showing chemical structures and schemes (e.g. cartoon of the core and/or shell). I fail to see the primary purpose of the paper and how the collection of disconnected results correlate to tell that story. Even the title is misleading – why is naphthalene anhydride even mentioned?
Specific comments
1. Lines 221 – 223: How can “The fluorescent polymer PAN with AN amount of 0.1 % was firstly synthesized via the above copolymerization without M-t-BMMs” be synthesized? There are no polymerizable vinyl groups on AN. Also, if used, the fluorescent polymer should be fully characterized (confirmation that AN is covalently bound to the polymer).
2. Line 228: “A moderate amount …” is not appropriate for an experimental section. Please disclose the masses used.
3. The XRD characterization (pgs. 8 – 9), are presented as a results and discussion, making it difficult to understand how the quantitative results (surface area, etc.) support the conclusion the authors are trying to make. Much of the information is provided in the SI, which makes it seem irrelevant to the “punchline” of the paper.
4. The characterization is not well presented and sloppy. For example, Figure S4 (used to justify the presence of AN via the extremely small, and not AN-dependent, 780 cm-1 peak) does not have curves referenced in the figure caption (h and higher).
5. The 3-nm wavelength shift shown in Figure 3A does not add value. Also, the discussion in lines 476 – 484 confuses the solvent-dependent emission with monomeric and aggregated states. They are two completely different phenomena.
6. The chemical structure shown in Figure 3D is not at all correct – bond angles are sloppily drawn; what are the formal charges on the two nitrogens; and why does the caption say “illustration of two chromophores?” I see an elaboration of this on page 13, but the model is not well-described. This is a complicated kinetic phenomenon, where the rates of both ground- and excited-state equilibria must be considered. This particular part of the paper (fluorescence lifetimes) is an excellent example of a potentially interesting result that is not necessarily relevant to the purpose of the work, but put in just because the authors had the data.
7. I believe the authors are using the results/discussion on pages 11 – 12 to state that there is AN aggregation at a specific loading. There are so many factors effecting fluorescence that the qualitative changes depicted in the figures could be due to a number of factors. It is also surprising that aggregation would be operative at such a small AN loading. The authors really need to consider an alternative way of characterizing the polymer used for the shell.
8. … the remainder of the paper continues to be a collection of results that are not well-organized or presented. It is simply a collection of results that are not coherently interpreted. In the absence of well-organized tables, clear presentation of results, and a concise discussion that connects all of the results, it is a very difficult piece of work to read and review. The heavy reliance on Supporting Information to show more data does not add value, and very much weakens the manuscript. The authors should consider carefully how they can reorganize and better correlate the different sets of results they have.
Author Response
Responses to the comments of Reviewer 2#

Reviewer 3 Report
I find the results of this study are well presented throughout and would be of interest to the readership of Polymers. I would recommend the article undergoes extensive proof-reading prior to publication. Overuse of transitional adverbs makes the manuscript hard to follow. I recommend publication following the authors consideration of the following comments:
1) For the Materials and Methods I have concerns regarding the lack of detail provided. Including throughout Section 2.9 there is limited detail regarding how the samples were prepared for the techniques applied. Without further information it makes assessing the validity of the data challenging. Please see example queries below:
- L208: What was the vacuum pressure?
- L272: What passage number were the cells?
- L275: How long were the cells cultured for?
- L283: State plate reader instrument used.
- L313: For SEM analysis: Were the samples coated? Which detector was used? What was the working distance? What was the beam current? For TEM analysis what accelerating voltage was selected?
- L318: How many scans did you average over? And at what resolution?
2) L432: FT-IR is not surface sensitive. Therefore, I do not see how vinyl functionalisation on the surface of BMMs is the rationale for these bands. Also, I would recommend checking your band allocation again with the literature you have cited. Additionally further annotation and explanation is required for Figure S4 FT-IR.
3) In respect to Figure 2 do you have any lower mag images of M-BMMs and PAN@BMMs-I- 4520.1? How representative are these regions? It would have been helpful for the authors to have performed image analysis of the pores and included an in-set.
4) Please check formatting of Figure 3 and 8. Some line errors are present. Please also check figure resolutions.
5) Why wasn't statistical analysis performed? This would be very useful for data presented in figure 9. Without performing statistical analysis conclusion claims (L925) cannot be verified.
6) L874: This study is not "in-vivo"?
7) Figure 9 contains error bars. What was the n number? Did you take into consideration tissue culture plastic and control media fluorescence?
8) CLSM images in Figure 10 are too small. Recommend amending this figure.
Author Response
Responses to the comments of Reviewer 3#

Round 2
Reviewer 1 Report
see attached document.

Author Response
Response letter for Reviewer1#

Reviewer 2 Report
The addition of Scheme I has been very helpful to understand the synthetic details of the PAN@BMMs. The other revisions provided by the authors do help to clarify some of the details. However, the body of work is not well organized and is based on highly speculative details of seemingly uncorrelated results. For example, the authors spend much time discussing AN aggregation, presumably using the lifetime curves and values from the biexponential fits without ever showing any fits. Much of the data collected in Table 1 is not described in the text (d?), and uncertainties are not included to support that e.g. 1.86 is different number than 2.02. The support for the hydrogen-bonded AN (shown in Scheme I cartoon) is not strongly provided by the results in the manuscript. Moreover, there is no quantification of what the AN “loading” actually is. The AN “loading” is misrepresented as the AN/monomer ratio which is not necessarily what is in the actual nanoparticle with a polymerized shell.
Specific comments like those described in the previous paragraph are too numerous to list. I suggest the authors think carefully about the “roadmap” of how each type of experiment and result feeds into the ultimate goal of understanding delivery function of the materials. I would also suggest that, once done, the cytotoxicity and imaging results be reinterpreted in the context of a more detailed understanding and characterization of the materials.
Author Response
Response letter for Reviewer 2#

Reviewer 3 Report
I would recommend this article is accepted with minor corrections.
1) Please could the authors state if their samples were coated or not prior to SEM analysis.
2) I would still dispute the surface sensitivity of ATR-FTR. I would expect ATR-FTIR to have an information depth of around 1-2 micron. Therefore, surface related statements the authors make (eg line 460 etc) cannot be confirmed by this method alone. I would recommend the use of XPS.
Author Response
Response letter for Reviewer 3#

Round 3
Reviewer 1 Report
see attachment

Author Response
Response to Reviewer 1#
